# Study protocol for development and validation of a single tool to assess risks of stroke, diabetes mellitus, myocardial infarction and dementia: DemNCD-Risk

Scherazad Kootar ![ORCID],[1,2] Md Hamidul Huque ![ORCID],[1,2] Kim M Kiely ![ORCID],[1,2] Craig S Anderson ![ORCID],[3,4] Louisa Jorm ![ORCID],[5] Miia Kivipelto ![ORCID],[6] Nicola T Lautenschlager ![ORCID],[7,8] Fiona Matthews ![ORCID],[9] Jonathan E Shaw ![ORCID],[10] Rachel A Whitmer,[11] Ruth Peters ![ORCID],[12] Kaarin J Anstey ![ORCID][1,2]

## ABSTRACT

**Introduction** Current efforts to reduce dementia focus on prevention and risk reduction by targeting modifiable risk factors. As dementia and cardiometabolic non-communicable diseases (NCDs) share risk factors, a single risk-estimating tool for dementia and multiple NCDs could be cost-effective and facilitate concurrent assessments as compared with a conventional single approach. The aim of this study is to develop and validate a new risk tool that estimates an individual's risk of developing dementia and other NCDs including diabetes mellitus, stroke and myocardial infarction. Once validated, it could be used by the public and general practitioners.

**Methods and analysis** Ten high-quality cohort studies from multiple countries were identified, which met eligibility criteria, including large representative samples, long-term follow-up, data on clinical diagnoses of dementia and NCDs, recognised modifiable risk factors for the four NCDs and mortality data. Pooled harmonised data from the cohorts will be used, with 65% randomly allocated for development of the predictive model and 35% for testing. Predictors include sociodemographic characteristics, general health risk factors and lifestyle/behavioural risk factors. A subdistribution hazard model will assess the risk factors' contribution to the outcome, adjusting for competing mortality risks. Point-based scoring algorithms will be built using predictor weights, internally validated and the discriminative ability and calibration of the model will be assessed for the outcomes. Sensitivity analyses will include recalculating risk scores using logistic regression.

**Ethics and dissemination** Ethics approval is provided by the University of New South Wales Human Research Ethics Committee (UNSW HREC; protocol numbers HC200515, HC3413). All data are deidentified and securely stored on servers at Neuroscience Research Australia. Study findings will be presented at conferences and published in peer-reviewed journals. The tool will be accessible as a public health resource. Knowledge translation and implementation work will explore strategies to apply the tool in clinical practice.

## INTRODUCTION

Non-communicable diseases (NCDs) are a global health issue leading to around 41 million deaths each year. Among these, cardiovascular diseases (CVD) account for most deaths (17.9 million), followed by cancer (9.3 million), respiratory illness (4.1 million) and diabetes (1.5 million).[1] According to the Centers for Disease Control and Prevention, USA, dementia ranked seventh for mortality in 2020, alongside the above diseases partially due to global population ageing.[2] Dementia is the largest contributor to the non-fatal burden of disease among older adults in Australia.[3] The Global Burden of Diseases, Injuries and Risk Factors[4] reports the estimated global number of people with dementia increasing from 57 million in 2019 to 153 million in 2050. This increase in the number of people with dementia will negatively impact on individuals, families, healthcare systems and society. Given that there is no cure or effective treatment to stop the progression of dementia, the field has shifted focus towards prevention and risk reduction to reduce the burden on health and social care systems. Evidence on modifiable risk factors for dementia has been comprehensively synthesised into recent

umbrella reviews,[5] [6] a Lancet Commission[7] [8] and WHO Guidelines.[9] These reviews show that many NCDs are also risk factors for dementia, including diabetes mellitus (DM), stroke and other CVD.[5] [6] In addition, these four NCDs also share common social and lifestyle related risk factors.[10] Given the overlap in risk between dementia and other NCDs, the development of common risk reduction strategies may be more cost-effective and efficient, than conventionally approaching each disease individually.

### STRENGTHS AND LIMITATIONS OF THIS STUDY

⇒ This international and multicohort collaboration draws on data from multiple countries including Australia, Netherlands, Singapore, UK and USA.
⇒ To our knowledge, this will be one of the first validated risk-prediction tools for multiple non-communicable diseases, including dementia, that accounts for competing risks and draws on data sets from diverse geographic regions.
⇒ External validation will be required to assess generalisability to under-represented communities, low-resource settings and other populations not included in the contributing studies.
⇒ We rely on post hoc harmonisation of risk factors, and not all risk factor measures are directly comparable across studies.

Risk assessments for NCDs are widely used in clinical practice, research and policymaking.[11–14] The Framingham Risk Score, a well-known example of a health risk assessment for CVD risk, was adapted to calculate the absolute CVD risk and is used in the clinical practice guidelines and recommendations of many countries to guide treatment for the primary prevention of CVD.[15] Risk assessments primarily identify individuals at high risk of developing the disease and determine preventive measures for reducing risk. They are also used to evaluate risk reduction interventions[15] [16] and population modelling of risk levels.[17] [18] Currently, there are numerous risk tools available for dementia,[19] DM,[20] heart disease[21] and stroke.[22] Examples of scoring systems used to predict dementia include the Cardiovascular Risk Factors, Aging, and Incidence of Dementia (CAIDE), the ANU-Alzheimer's Disease Risk Index (ANU-ADRI) and the Cognitive Health and Dementia Risk Reduction (CogDrisk).[23–26] Similarly, for the cardiometabolic field, there are Framingham risk scores for CVD (10-year and 30-year risk),[27] [28] the Australian type 2 Diabetes Risk Assessment tool (AUSDRISK),[29] the FINDRISC Diabetes risk tool,[30] the Cambridge Diabetes Risk Score,[31] the Framingham diabetes risk score to assess risk of DM[29] [32] and the Framingham stroke risk score.[22] Variables such as lack of physical activity, unhealthy diet, smoking, hypertension and obesity are relevant to dementia and overlap with risk factors for DM and CVDs. To our knowledge, there is no single risk prediction tool for NCDs that includes dementia. Building such a tool that accounts for substantial overlap in risk factors and interactions between NCDs may be cost-effective and save clinical assessment time as compared with assessing each disease individually.

To address the need for a population-based prediction tool for incidence of multiple NCDs including dementia, we propose the development and validation of the DemNCD-Risk tool. This risk tool will provide a combined estimate for an individual's risk of developing dementia, type 2 DM, stroke and myocardial infarction (MI). We anticipate that a multi-domain risk tool that accounts for shared and competing risk will be more likely to be used in primary care as it would reduce assessment time. This study protocol outlines the analytical approach used to develop and validate our risk tool.

## METHODS AND ANALYSIS
### Study design
Prior to commencing the study, we determined the minimum sample size required and types of cohorts that would be suitable. Inclusion criteria were population-based cohort studies of mid-life and/or older age participants, with at least 5 years follow-up (exception being Cognitive Function and Ageing Studies (CFAS)-II with follow-up time of 2 years), having clinical diagnosis of dementia and other NCDs and the availability of the majority of risk factors used in current prediction tools. We included CFAS-II due to its similarity to CFAS-I in terms of being based in England and having a comparable population and characteristics. Moreover, the availability of accessible data of CFAS-II along with CFAS-I supported our rationale for including this study in our analysis.

We identified potential data sets through searches of consortia websites, databases, word of mouth and consultation with experts. We evaluated cohorts in terms of the outcome measures, including studies with the availability of clinical diagnosis of dementia and other NCDs, risk factors and measures (which will also be referred to as predictors in the methodology), length of follow-up time, sample size requirements and the availability of data from the study custodians. We shortlisted 10 cohort studies that met our criteria: the Cardiovascular Health Study (CHS),[33] the Framingham Heart Study (FHS),[34–36] the MRC CFAS-I[37] [38] and CFAS-II[39], the Sydney Memory and Aging Study (MAS),[40] the Maastricht Aging Study (MAAS),[41] [42] the Health and Retirement Study- Aging, Demographics and Memory Study (HRS ADAMS),[43–45] the Memory and Aging Project,[46] the Atherosclerosis Risk in Communities (ARIC)[47] [48] and the Singapore Longitudinal Ageing Studies-I (SLAS-I).[49] [50] Key characteristics for each study are available in table 1. Ethics approval was obtained from the UNSW Human Research Ethics Committee (approval number HC00515) and approval from the data custodians was obtained to access the data sets.

We report the study protocol following the Transparent Reporting of a multivariable prediction model for Individual Prognosis or Diagnosis statement for prediction studies.

### Patient and public involvement
Patients and public were not involved in the design, conduct, reporting or dissemination plans of our research.

### Participants
Individual data will be pooled from the 10 cohort studies (age 44–110 years). Participants from these cohorts who have dementia at baseline will be excluded.

### Outcomes
The primary objective of the current study is to develop a validated risk assessment tool to predict four different NCDs, including dementia, MI, stroke and DM. We chose to predict for MI instead of coronary heart disease (CHD) as only CHS and FHS had a composite variable for CHD, which was adjudicated by a subcommittee while MI was reported in all studies. These NCDs were chosen as they are highly prevalent, related, and they share common modifiable risk factors that are often assessed and managed in primary care. Clinical diagnosis of each of these outcomes will be used (where

**Table 1** Characteristics of contributing studies

| Study | Baseline year | Age range at baseline | Number of individuals at baseline | Males (n) and Females (n) | Number of waves and year | Follow-up time | Country |
|---|---|---|---|---|---|---|---|
| Maastricht Aging Study (MAAS) | 1992 | 24–81 years | 1823 | Women=910, Men=913 | Baseline (1992–1996), Wave 1 (1996–1999), Wave 2 (1999–2002), Wave 3 (2002–2005), Wave 4 (2005–2008), Wave 5 (2018–2020) | 25 years | The Netherlands |
| The Framingham Heart Study (FHS) | 1948 | 30–62 years | 5209 | Women=2873, Men=2336 | Wave 1 (1948–1953) to Wave 32 (2012–2014) | 61 years | USA |
| Atherosclerosis Risk in Communities (ARIC) | 1985 | 44–66 years | 15028 | Women=8198, Men=6830 | Wave 1 (1987–1989), Wave 2 (1990–1992), Wave 3 (1993–1995), Wave 4 (1996–1998), Wave 5 (2011–2013), Wave 6 (2015–2016), Wave 7 (2017–2018), Wave 8 (2019–2020) | 31 years | USA |
| Health and Retirement Study - Aging, Demographics and Memory Study (HRS ADAMS) | 1992 | 70 years and above | 856 | Women=501, Men=355 | Wave 1 (2002–2005), Wave 2 (2006–2008), Wave 3 (2008–2009) | 8 years | USA |
| The Singapore Longitudinal Ageing Study (SLAS I) | 2003 | 55 years and above | 2567 | Women=1666, Men=901 | Wave 1 (2003–2005) and Wave 2 (2007–2009) | 6 years | Singapore |
| The Cardiovascular Health Study (CHS) | 1991–1994 | 65–100 years | 5888 | Women=3393, Men=2495 | Wave 1 (1991–1994) Final follow-up (1999–2000) | 8 years | USA |
| Cognitive Function and Ageing Study (CFAS-I) | 1989–1991 | 65–105 years and above | 13004 | Women=7847, Men=5157 | Wave 1 (1991–1994), Wave 2 (1991–1995), Wave 3 (1992–1996), Wave 4 (1994–1998), Wave 5 (1997–2001) | 10 years | UK |
| Cognitive Function and Ageing Study (CFAS-II) | 2008 | 65–100 years and above | 7762 | Women=4228, Men=3534 | Wave 1 (2008–2011) Wave 2 (2011–2013) | 2 years | UK |
| Sydney Memory and Aging Study (MAS) | 2005 | 70–90 years | 1037 | Women=572, Men=465 | Wave 1 (2005–2007), Wave 2 (2007–2009), Wave 3 (2009–2011), Wave 4 (2011–2013), Wave 5 (2013–2016), Wave 6 (2016–2018), Wave 7 (2018 onwards) | 12 years | Australia |
| Rush Memory and Aging Project (MAP) | 1997 | 55 years and above | 2192 | Women=1610, Men=582 | Wave 1 (1998) to Wave 22 (2019 and onwards) | 22 years | USA |

available), however, self-reports of diagnosis may also be used for stroke, diabetes and MI. Detailed descriptions of each of these outcomes are presented and described in table 2:

**Table 2** Considerations for defining the outcomes

| Number | Outcome | Considerations |
|---|---|---|
| 1 | Dementia | Clinical diagnosis of dementia |
| 2 | Stroke | Clinical diagnosis of stroke, if not available then self-reported stroke |
| 3 | Diabetes | Clinical diagnosis of diabetes or derived variable or lab results, if not available then self-reported diabetes |
| 4 | MI | Clinical diagnosis of MI, if not available then self-reported MI |

MI, myocardial infarction.

### Dementia

All included studies diagnosed dementia through the Diagnostic and Statistical Manual of Mental Disorders (III-R, IV criteria) or International Classification of Diseases codes. We chose to study all-type dementia instead of specific types for two reasons: one, as there are no tools available to understand the different dementia pathologies to predict dementia types; and two, as we intend to build a tool, which is simple and facilitates public participation. Algorithms were used for diagnosis of dementia in ARIC[48] and CFAS.[51]

### Stroke

For the ARIC study, a diagnostic algorithm created variable that classifies stroke events based on data from reports will be used to define stroke. For the CHS, special cerebrovascular subcommittees defined the criteria for stroke and TIA,[52] while in the FHS, stroke cases were reviewed by

neurologists. Where clinical diagnosis is not available, for example, CFAS I and II, Sydney MAS, SLAS-I and MAAS, and HRS ADAMS self-reported stroke will be used.

## Diabetes mellitus

Similar to stroke, clinical diagnosis of DM (including type of diabetes where information is available), derived variables or lab results, will be preferred over self-reported DM. Derived variables for diabetes based on WHO guidelines or American Diabetes Association guidelines are available in the CHS while self-reported type 2 diabetes will be considered for other studies.

## Myocardial infarction

All the studies report self-reported heart attacks.

## Death

Date/year of death or age at death are available in all studies.

## Predictors

We identified model predictors in three stages. First, we identify predictors from the latest seminal systematic reviews for the four outcomes,[5 6 10 53–57] the Lancet Commission on dementia[7 8] and the WHO Guidelines for evidence on risk factors and risk reduction of cognitive decline and dementia.[9] Second, predictors from commonly used risk tools were collated into a long list. For this, we referred to the ANU-ADRI, LIfestyle for BRAin Health (LIBRA) and CAIDE for dementia predictors, the Framingham Heart Study CHD risk tool (10-year risk for MI or coronary death), Framingham risk score for CVD (10 year and 30 year), CVD check for MI predictors, the AUSDRISK and Framingham diabetes risk score for diabetes predictors and the Framingham stroke risk score for stroke predictors (refer table 3). Finally, the long list of potential predictors was then reviewed and voted by eight subject matter experts who had previous experience with population-based prediction models.[24 29] Ethics approval was obtained from UNSW Human Research Ethics Committee (protocol number HC3413). We received vote from six experts for dementia predictors, three each for predictors of stroke, diabetes and MI. We selected all of the predictors who were nominated by any two of the reviewers and were available in the data set. In total, we identified 18 predictors for stroke, 15 for diabetes, 18 for CHD and 23 for dementia. Predictors were mainly extracted from the baseline examination. In cases where this was not available, predictors were selected from the first wave in which they were measured.

Candidate predictors considered for inclusion in the predictive model (definitions are provided in table 4):

► *Demographic factors*: age, sex, education level, ethnicity/country of birth, socioeconomic status and family history.
► *Lifestyle risk/protective factors*: smoking, harmful alcohol consumption, fish intake/healthy diet, insufficient physical activity, social isolation, lack of cognitive engagement and sleep disturbance.
► *Medical risk factors*: mid-life obesity, hypertension, hypercholesterolemia, triglycerides, low and high-density lipoproteins, late-life atrial fibrillation, chronic kidney disease, high blood glucose, urinary albumin and protein, left ventricular hypertrophy, depression, traumatic brain injury, high homocysteine, stress, orthostatic hypotension and hearing loss.

Table 4 reports the risk factors and their definitions.

## Sample size

We assumed that the proposed risk algorithms that generate risk scores for dementia and other outcomes will have sensitivity between 70% and 90% across different cohorts as the prevalence of disease will vary across cohorts. Assuming global prevalence of dementia at age 60 and older is 5%, we estimated that nine studies with sample sizes between 1041 and 2165 will be needed to obtain a sensitivity of 70% and 90%, respectively, with 10% precision and 5% level of significance with Bonferroni correction. We focused on dementia for sample size calculations, as among all the outcomes considered for this study, dementia has the lowest prevalence. Therefore, the minimum sample size for dementia will be sufficient to provide required statistical power for other outcomes.

## Statistical analysis methods
### Analysis plan

We will pool harmonised individual data from the 10 studies to develop a risk tool that provides long-term risk estimates for diabetes, stroke, MI and dementia. Machine learning methods will be used to develop and validate the risk tools. Specifically, we will use 65% of the pooled data in training the model, and the remaining 35% of the data will serve as test data for internal validation in order to ensure good coverage of the outcomes in the validation sample.[58] We plan to validate the tool on additional data sets in future work. As our objective is to build a tool that predicts long-term health outcomes, we will use competing risk survival models (Fine and Gray subdistribution hazard model[59]). When predicting outcomes over long periods of time, competing risks should account for risk of death from non-disease-related causes as this risk increases over time and decedents are no longer at risk of a disease incidence. As the competing risk of mortality is more common in high-risk populations with a long duration of follow-up, this effect may be more pronounced in older cohorts than in younger or healthy cohorts. Moreover, this model allows one to estimate the effect of covariates on the cumulative incidence for the event of interest. In addition, such model needs to incorporate time-varying covariates or to include the factor of time. Since we are using different longitudinal cohorts, not all the time-varying covariates may be available in the longitudinal follow-up in every cohort. In this case, we will develop risk prediction models targeting mid-life (aged 45 to 65 years) and late-life (65 years onward). Predictor weights from this model for risk factor exposure and their interactions will be combined to build point-based scoring

**Table 3** List of predictors included in current risk tools for dementia, diabetes, myocardial infarction and stroke

| Risk factor category | Dementia risk tools (CAIDE, ANU ADRI, LIBRA) | Cardiovascular risk tools (Framingham Risk Tools- Hard Coronary Heart Disease, 10-year/30-year risk of CVD, CVD check) | Diabetes risk tools (AUSDRISK, Framingham risk score for diabetes) | Stroke risk tool (Framingham stroke risk score) |
|---|---|---|---|---|
| Demographic variables | | | | |
| Age and sex | Y | Y | Y | Y |
| Education | Y | – | – | – |
| Ethnicity | – | Y | Y | – |
| Parental/family history of disease | – | Y | Y | – |
| Medical risk factors | | | | |
| Hypertension/systolic BP | Y | Y | Y | Y |
| Use of antihypertensives | – | Y | Y | Y |
| High total cholesterol | Y | Y | – | – |
| HDL cholesterol | – | Y | Y | – |
| Obesity/waist circumference | Y | Y | Y | – |
| Diabetes | Y | Y | – | Y |
| Depression | Y | – | – | – |
| Traumatic brain injury | Y | – | – | – |
| High blood glucose levels | – | – | Y | – |
| Triglycerides | – | – | Y | – |
| Chronic kidney disease/renal dysfunction | Y | Y | – | – |
| Urine for microalbumin and protein | – | Y | – | – |
| Atrial fibrillation | – | Y | – | Y |
| Prior cardiovascular disease | Y | – | – | Y |
| Left ventricular hypertrophy | – | – | – | Y |
| Lifestyle and behavioural risk factors | | | | |
| Smoking | Y | Y | Y | Y |
| Physical inactivity | Y | Y | Y | – |
| Diet (fish intake, consumption of vegetables and fruits) | Y | Y | Y | – |
| Alcohol consumption | Y | Y | – | – |
| Social engagement | Y | – | – | – |
| Cognitive engagement | Y | – | – | – |
| Environmental exposure | | | | |
| Pesticide exposure | Y | – | – | – |

ANU-ADRI, Australian National University Alzheimer's Disease Risk Index; AUSDRISK, Australian type 2 Diabetes Risk Assessment tool; BP, blood pressure; CAIDE, Cardiovascular Risk Factors, Aging, and Incidence of Dementia; CVD, cardiovascular disease; LIBRA, LIfestyle and BRAin Health.

algorithms that provide individualised risk prediction.[60] The individual risk scores will be validated internally using the area under the curve (AUC).[61] Model calibration will also be carried out. We will estimate sensitivity and specificity of each risk algorithm based on a suitable cut-off on the receiver operating characteristic (ROC) curve.

### Data harmonisation and coding of predictors
Each study wave will be screened for the availability of risk factors and outcomes using the data dictionaries. Predictors and outcomes listed above will be checked for availability of the measure in each wave of the different datasets. Data harmonisation will be carried out as described previously[62] using the 'by fiat' method.[63] This method is developed using a common scoring system across studies to standardise the measure of the variables. In certain cases, for example, cognitive engagement, we will capture the total cognitive activity per study as compared with harmonising the variable across the studies as different items of cognitive activity are measured in each study. Detailed definitions and measurements of the predictors are presented in table 4.

**Table 4** Considerations for defining the candidate predictors

| Predictors | Types of predictors | Considerations |
|---|---|---|
| Demographic factors | | |
| Education | Categorical | Less than secondary vs secondary and vocational training vs tertiary |
| Ethnicity and country of birth | Categorical | Questions related to race/ethnicity and country of birth |
| Socioeconomic status | Categorical | Questions on occupation, income and accommodation |
| Family history | Binary | Family history of stroke, diabetes, dementia, MI |
| Lifestyle risk factors | | |
| Smoking status | Binary | Current vs former vs never |
| Alcohol consumption | Categorical | Abstinence vs drinker<br>Average number of drinks per drinking session<br>Heavy drinker vs non heavy drinker |
| Fish intake/healthy diet | Binary | Fish intake- more than twice per week<br>Fruit and vegetable intake (at least five servings of vegetables or three servings of vegetables and two servings of fruits) |
| Physical inactivity | Binary | Total time per week of physical activity—more than 500 MET per week—physically active vs inactive<br>Sedentary—yes vs no |
| Social isolation | Binary | Loneliness vs not lonely |
| Cognitive engagement | Categorical | Decided to use the total cognitive activity as reported by each study and then to divided into tertiles/quartiles—depending on each study to get a cut-off for cognitive inactivity. |
| Sleep disturbance | Binary | Account for those with at least two symptoms of the three self-reported sleep problems: trouble falling asleep, wake up several times at night, wake up too early |
| Medical risk factors | | |
| BMI categories | Categorical | WHO classifications—underweight (BMI<18.5) vs normal (BMI 18.5–24.9) vs overweight (BMI 25–29.9) vs obese (BMI>30) |
| Hypertension status | Binary | Yes—blood pressure average systolic≥140 mm Hg OR average diastolic≥90 OR history of hypertension OR currently using antihypertensives vs no |
| High cholesterol and triglycerides | Binary | High cholesterol (self-reported/lifetime history or total cholesterol>6.18 mmol/L) |
| | | High triglycerides (>2.3 mmol/L triglycerides) |
| | | High cholesterol or high triglycerides (self-reported) |
| Low-density lipoproteins | Binary | >4.1 mmol/L of LDL cut-off is high LDL |
| High-density lipoprotein | Binary | <1 mmol/L (for men) and<1.3 mmol/L (for women) of HDL cut-off is low HDL |
| Atrial fibrillation | Binary | AF measured by ECG or self-reported history of AF |
| Chronic kidney disease | Binary | First preference clinical diagnosis of kidney disease, if not use self-reported history of kidney disease |
| High blood glucose | Binary | To consider borderline or pre-diabetes cut-off for blood glucose—100–125 mg/dL or 5.6–7.0 mmol/L |
| Urinary microalbumin and protein | Categorical | Lab reports of urine microalbumin and protein levels |
| Left ventricular hypertrophy | Binary | Yes or no for LVH as defined by study |
| Depression | Binary | Self-reported depression symptoms measured by respective scales of depression used by the respective studies |
| Traumatic brain injury | Binary | Self-reported history of brain injury/head injury/knock out with or without loss of consciousness |
| High homocysteine | Binary | Lab reports of homocysteine levels |
| Stress | Binary | Self-reported stress measures |
| Orthostatic hypotension | Binary | Use self-reported orthostatic hypotension or calculated variable |
| Hearing loss | Binary | To categorise in three groups—no hearing loss, has hearing loss and wears hearing aid, has hearing loss but does not wear a hearing aid. |

AF, atrial fibrillation; BMI, body mass index; HDL, high-density lipoprotein; LDL, low-density lipoprotein; LVH, left ventricular hypertrophy; MET, metabolic equivalent of task; MI, myocardial infarction.;

We will use national/international guidelines where possible, for example; 150 min of moderate and vigorous physical activity per week, which is equivalent to >500 MET minutes per week of physical activity,[64] five or more servings of vegetables or three to five servings of fruit and vegetables and more than two servings of fish per week will be used to represent a low-risk diet (as measured by the American Heart Association's (AHA) Life's Simple seven guidelines,[65] alcohol consumption of not more than 10 standard drinks (one standard drink=10 g of pure alcohol) per week as recommended by the Australian guidelines.[66] For biological markers, we will use cut-offs informed by national and international clinical practice guidelines and recommendations from national healthcare providers, for example, the Royal Australia College of General Practitioners.[13 67 68] Cut-offs to be used: high-density lipoprotein (<1 mmol/L for men and <1.3 mmol/L for women), low-density lipoprotein (>4.1 mmol/L), high cholesterol (>6.18 mmol/L) and triglycerides (>2.3 mmol/L). We will choose 'high' instead of 'borderline high' as the most robust category for the prediction of disease outcomes. Self-report stress measures will be used, clinical thresholds for high homocysteine, urinary albumin and protein if reported in articles from the cohort studies to be included.

### Missing data

Missing data are inevitable in studies with long follow-up and may lead to bias. Variables with missing data will be imputed through multiple imputation across all studies.[69] Specifically, a multivariate normal imputation or imputation by fully conditional specification, where appropriate, will be employed to impute missing data. Compatibility/congeniality between imputation model and analysis model will be ensured. The number of imputations will be determined according to the current guidelines.[70]

### Model specification

Following multiple imputation, competing risk regression with available covariates (candidate covariates that are available in the data set) and death as a competing event will be modelled for each outcome, stratified by sex. Robust sandwich estimation of the variance will be obtained. All the cohorts under consideration are likely to be heterogeneous in terms of sample sizes and underlying population characteristics, that is, outcome and covariate distribution. Moreover, running competing risk analysis with multiple imputed data sets on large number of data may cause the slow convergence of the program. Therefore, we will run the competing risk regression model in each of the datasets separately. As we plan to evaluate the competing risk model in individual data sets, we will not be able to create 10-year risk prediction estimates.

### Model estimation

Estimates of the regression models for multiple imputed models will be combined using Rubin's rule in each cohort. The resulting regression estimates will be combined across cohorts to get final estimates using meta-analysis. The regression coefficients estimate from meta-analysis will then be used to calculate the point-based risk score for men and women separately,[26] which we will then test with the testing data sets for prediction of each of the outcomes.[60] All the analyses will be carried out using Stata V.16.0.

### Assessment of predictive performance

AUC, sensitivity and specificity of the risk scores on the test data set will be obtained to assess the discriminatory power. We will also assess the calibration of the risk algorithms using calibration plots and Hosmer-Lemeshow statistics.[71]

### Model presentation

The final regression coefficients (95% CI) for dementia, stroke, diabetes and MI, along with the points corresponding to the final algorithm, will be made available.

### Analysis beyond initial model development

In addition to the complex survival model, we will also calculate risk scores from the cohorts under consideration using logistic regression to test for the effect of follow-up time on risk scores. Specifically, we will be modelling binary outcomes with all the predictors ignoring follow-up time for the initial model. The risk score will then be calculated using the regression coefficients from the logistic model using a similar methodology.

In addition to the above model with all the risk factors, we are also planning to build a model based on the best subset of predictors. Specifically, for each of outcome, we are planning to perform backward elimination (p<0.1). The risk scores for each outcome will then be calculated using the regression coefficients obtained from the most parsimonious model. The AUC (95% CI) of the most parsimonious model will then be compared with the AUC obtained from the full model.

### Translation of the analyses into a risk assessment for implementation

To generate information on risk of dementia, DM, stroke and MI using a single assessment, we will create a single set of questions that encompass all the risk factors, so that the patient experience is a single assessment. Algorithms will be applied to this tool to generate advice and risk estimates for multiple outcomes. Ideally, the analyses of the cohort studies will provide a risk algorithm for each outcome. However, this may be supplemented by existing risk algorithms, if the results of the proposed analyses are not sufficient. Where possible, risk factors will be assessed as measured in the original cohort studies. Where the quality of measures is mixed or of suboptimal quality, we will identify validated measures. For example, with respect to the measurement of insomnia, some studies use clinical sleep scales and others include a study-specific self-reported item. We will use a standardised and validated instrument for diagnosis of insomnia in the assessment tool. We anticipate that two assessment tools

will be created: one for the clinicians and another for researchers and the public. This is due to the fact that clinicians have limited consultation time; their questionnaire will be shorter as compared with the public one.

## Limitations

Our study has some limitations. Although using multiple cohorts to develop the tool is an improvement compared with a single cohort, our cohorts are drawn mainly from high-income countries and with most of them being mainly Caucasian due to the availability of these data sets with the required information. This will limit the generalisability of the tool. We are unable to take account of the differences in treatment and care that will have influenced participants in different cohorts.

However, as the tool will be developed and validated from data pooled from America, Asia, Australia and Europe, we have some diversity within our sample. As the tool will be validated on the same sample as the derivation cohort, external validation of the tool will be desirable after study completion. We are unable to account for cause of death in the analysis as most of the studies lack this information.

Moreover, as we will be using secondary data from various cohort studies to estimate the risk of dementia, stroke, MI and diabetes; therefore, the specific types of diagnosis of above conditions may not be available in all studies (eg, ischaemic vs haemorrhagic stroke, type-I vs type-II diabetes etc). In addition, the method of diagnosis may vary across different studies. This may cause potential biases. To assess the impact of this and other cohort-specific biases, we will study the cohort-specific prediction in addition to aggregate predictions for all cohorts.

## ETHICS AND MODEL DISSEMINATION

Ethics approval is provided by the University of New South Wales Human Research Ethics Committee (UNSW HREC; protocol numbers HC200515, HC3413). All data are deidentified and stored on a secure server at Neuroscience Research Australia. Study findings will be presented at relevant national and international conferences and published in peer review journals. The tool will be made available as a public health resource. Knowledge translation and implementation work will explore strategies to apply the tool in clinical practice.

**Author affiliations**
[1]Neuroscience Research Australia, Randwick, New South Wales, Australia
[2]School of Psychology, University of New South Wales, Sydney, New South Wales, Australia
[3]The George Institute for Global Health, George Institute for Global Health, Newtown, New South Wales, Australia
[4]Faculty of Medicine, University of New South Wales, Kensington, NSW, Australia
[5]Centre for Big Data Research in Health, University of New South Wales, Randwick, New South Wales, Australia
[6]Division of Geriatric Epidemiology, Karolinska Institutet, Stockholm, Sweden
[7]Academic Unit of Psychiatry of Old Age, Department of Psychiatry, The University of Melbourne, Parkville, Victoria, Australia
[8]Older Adult Mental Health Program, Royal Melbourne Hospital Mental Health Service, Parkville, Victoria, Australia
[9]Population Health Sciences Institute, Newcastle University, Newcastle upon Tyne, UK
[10]Clinical and Population Health, Baker Heart and Diabetes Institute, Melbourne, Victoria, Australia
[11]University of California Davis, Davis, California, USA
[12]University of New South Wales, Sydney, New South Wales, Australia

**Contributors** KJA is the lead investigator of this study and oversaw the project development, co-developed the original project idea; contributed towards developing the analysis plan, study design including identifying various cohort studies, applied to various studies and got approval, and planned the harmonisation of outcomes and predictors. Critically reviewed and contributed to the protocol and approved the final draft. RP co-developed the original project idea; contributed towards developing the analysis plan, study design including identifying various cohort studies and planned the harmonisation of outcomes and predictors. Critically reviewed and contributed to the protocol and approved the final draft. SK contributed towards developing the analysis plan, study design including identifying various cohort studies, applied to various studies and got approval and planned the harmonisation of outcomes and predictors. Reviewed the literature, conducted the survey on subject matter experts for selecting predictors and prepared the first draft of the manuscript. Critically reviewed and contributed to the protocol and approved the final draft. HH contributed towards developing the analysis plan, study design including identifying various cohort studies, planning the harmonisation of outcomes and predictors and drafted the method section. Critically reviewed and contributed to the protocol and approved the final draft. KK contributed towards developing the analysis plan, study design including identifying various cohort studies, and planning the harmonisation of outcomes and predictors. Critically reviewed and contributed to the protocol and approved the final draft. CA, LJ, NL, FM, JES, RAW, MK provided inputs to the analysis plan, critically reviewed and contributed to the protocol and approved the final draft.

**Funding** This research was funded by a NeuRA Grant to KJA, the NHMRC Dementia Collaborative Research Centre, and NHMRC GNT1171279. KJA is funded by the Australian Research Council Fellowship FL190100011, SK and HH are part funded by NHMRC GNT1171279. KK is partly funded by ARC CE170100005. JES is supported by NHMRC Investigator Grant APP1173952. CSA holds research grants and a Senior Investigator Fellowship from the NHMRC.

**Competing interests** JES has received honoraria for scientific advisory, lectures, and clinical research from Pfizer; Roche; Zuellig Pharma; Astra Zeneca; Sanofi; Novo Nordisk; MSD; Eli Lilly; Abbott; Mylan; Boehringer Ingelheim. CSA has received grants from Takeda, Penumbra and Credit Pharma awarded to his institution outside of this work.

**Patient and public involvement** Patients and/or the public were not involved in the design, or conduct, or reporting, or dissemination plans of this research.

**Patient consent for publication** Not applicable.

**Provenance and peer review** Not commissioned; externally peer reviewed.

**ORCID iDs**
Scherazad Kootar http://orcid.org/0000-0001-5496-3281
Md Hamidul Huque http://orcid.org/0000-0002-5605-3801
Kim M Kiely http://orcid.org/0000-0001-5876-3201
Craig S Anderson http://orcid.org/0000-0002-7248-4863
Louisa Jorm http://orcid.org/0000-0003-0390-661X
Miia Kivipelto http://orcid.org/0000-0003-0992-3875
Nicola T Lautenschlager http://orcid.org/0000-0003-4850-7794
Fiona Matthews http://orcid.org/0000-0002-1728-2388
Jonathan E Shaw http://orcid.org/0000-0002-6187-2203
Ruth Peters http://orcid.org/0000-0003-0148-3617
Kaarin J Anstey http://orcid.org/0000-0002-9706-9316

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
