## [Reviewer comments · BMJ Open]

ARTICLE DETAILS

TITLE (PROVISIONAL)	Study protocol for development and validation of a single tool to assess risks of stroke, diabetes mellitus, myocardial infarction, and dementia: DemNCD-Risk
AUTHORS	Kootar, Scherazad; Huque, Md Hamidul; Kiely, Kim; Anderson, Craig; Jorm, Louisa; Kivipelto, Miia; Lautenschlager, Nicola; Matthews, Fiona; Shaw, Jonathan; Whitmer, Rachel; Peters, Ruth; Anstey, Kaarin

VERSION 1 – REVIEW

REVIEWER	Jiang, Yongjun Guangzhou Medical University Second Affiliated Hospital
REVIEW RETURNED	25-Jul-2023

GENERAL COMMENTS	1. Is the stroke defined as ischemic or hemorrhagic? 2. Any steering committee would review the diagnosis or just derived from the record? Because the data from each individual cohort might vary from the different regions or decades.
--

REVIEWER	Harun, Sabariah Noor Universiti Sains Malaysia
REVIEW RETURNED	07-Aug-2023

GENERAL COMMENTS	1. How do you reduce the bias in predicting the dementia as different diagnostic criteria/algorithm used to confirm the disease. Will you test the different diagnostic criteria and types of dementia on the prediction? At least sensitivity analysis to confirm that it would not affect the prediction. 2. What type of stroke do you focus in this predictive tool? 3. Page 10, line 34- ranking the potential variables Any specific method or criteria to rank the potential predictors? Good to show any rubric or criteria used to rank the variables. Besides this, would it be possible to test the effect of all predictors in the model since you are using AI approach. 4: Page 10, line 48-50 Predictors were extracted from the baseline examination or from the first wave in which they were measured 5. Please elaborate more about this. Presumably the predictive tool will be developed based on cross sectional data rather than longitudinal. Authors may need to consider the factors of time as all NCDs focused on the tool are diseases that are progressing with time. Thus, variables that time varying, time when the NCDs developed may need to be taken into account into the prediction.
--

	6. Page 10, line 53- candidate predictors It would be great if type of variables can be specified-either continuous, ordinal or binary. categorical 7. Page 11, line 30: Sample size Please provide evidence to support the sample size calculation used for the model. Besides, is there any number of data used for model development and model validation? How authors determine this. 8. Page 12, line 21 If this is the case, the current work may not produce validated tool as stated in the title. Internal validation is a must procedure to develop a model. 9. Page 12, line 28 Any specific duration of time to the event e.g., dementia. Will you specify the risk based on different duration? 10. Page 14: line 3 What type of stress measures will be incorporated into the model? 11. Page 15: Any specific method used for this meta regression. Will you use machine learning approach? As meta regression applying conventional methods- there are few factors need to be considered and explained such multicollinearity of the data and cofounding factors that may affect the prediction. Authors may need to explain how they will take this into account. 12. How the internal validation will be performed? 13. Any specific method used for this survival model development? 14. Page 16, line 47 What would be the difference between the two tools and why it should be different?
--	--

VERSION 1 – AUTHOR RESPONSE

Reviewer: 1

Dr. Yongjun Jiang, Guangzhou Medical University Second Affiliated Hospital

Comments to the Author:

Comment 1. Is the stroke defined as ischemic or hemorrhagic?

Response 1: As we are using secondary data from various studies to estimate the risk of stroke and most of these studies use self-reported history of stroke, the type of stroke is unavailable for most of the datasets in this study. We added this as a limitation in the manuscript.

Comment 2. Any steering committee would review the diagnosis or just derived from the record? Because the data from each individual cohort might vary from the different regions or decades.

Response 2: Thanks for your question. As we are using secondary data, we rely on the diagnosis ascertained from the studies. Yes, we acknowledge that the criteria used by committees to define outcomes may vary between studies and is a limitation in our study. We have added those limitations in the limitation section of the protocol paper.

Reviewer: 2

Dr. Sabariah Noor Harun, Universiti Sains Malaysia

Comments to the Author:

Comment 1. How do you reduce the bias in predicting the dementia as different diagnostic criteria/algorithm used to confirm the disease. Will you test the different diagnostic criteria and types of dementia on the prediction? At least sensitivity analysis to confirm that it would not affect the prediction.

Response 1: We have selected 10 different cohorts that have clinical diagnosis of dementia through the Diagnostic and Statistical manual of mental disorder (DSM-III, IV) criteria or other well established criteria. These criteria of dementia diagnosis includes geriatric Mental State–Automated Geriatric Examination for Computer Assisted Taxonomy (GMS-AGECAT), criteria of National Institute of Neurological and Communicative Disorders and Stroke and the Alzheimer’s disease and Related Disorders Association. Participants in each of the datasets were evaluated for dementia by neurological and neuropsychological examination. Dementia diagnosis in each of the studies were well established.

We agree with the reviewer that there might be some difference in the diagnostic criteria. However, the detailed neurological and neuropsychological exam to validate the diagnosis criteria is unavailable to us. Moreover, the objective of the research does not include testing the impact of different diagnostic criteria of dementia, and many meta-analytic studies on risk factors for dementia have used the same approach as us. We assume that dementia diagnosis in different studies captures the true underlying dementia diagnosis. Therefore, we aimed to build a prediction model by aggregating information from all of these studies. In addition to validating the model in the combined validation sample, we will validate our model using each of the component of the validation sample to explore any substantial difference across studies. The potential bias associated with the differential diagnosis (if any) is beyond the scope of our current research but we have now added as a limitation in the limitation section.

Comment 2. What type of stroke do you focus in this predictive tool?

Response 2: We are limited by the availability of the data related to stroke in our project. As reported in page 8, line 41-42, most of the cohorts did not include classification of strokes rather self-report of any stroke. Hence, we aimed to predict overall stroke risk due to any type of stroke in this predictive tool.

Comment 3. Page 10, line 34- ranking the potential variables

Any specific method or criteria to rank the potential predictors? Good to show any rubric or criteria used to rank the variables. Besides this, would it be possible to test the effect of all predictors in the model since you are using AI approach.

Response 3: Thanks for your comment. Following your comments, we realized that the word “rank” was not quite right. Therefore, we have changed the word rank to vote. The current modified paragraph (page 9 line 19-23) reads as “Lastly, the long list of potential predictors was then reviewed and voted by eight subject matter experts who had previous experience with population-based prediction models (1, 2). Ethics approval was obtained from UNSW Human Research Ethics Committee (protocol number HC3413). We received vote from six experts for dementia predictors, three each for predictors of stroke, diabetes and myocardial infarction. We have selected all of the predictors that were nominated by any two of the reviewers and are available in the dataset.” Hope this is clear now. Yes, we will test the effect of all the potential predictors in the model to calculate the risk score as given in the model specification section of the manuscript.

Comment 4: Page 10, line 48-50

Predictors were extracted from the baseline examination or from the first wave in which they were measured. Please elaborate more about this. Presumably the predictive tool will be developed based on cross sectional data rather than longitudinal. Authors may need to consider the factors of time as all NCDs focused on the tool are diseases that are progressing with time. Thus, variables that time varying, time when the NCDs developed may need to be taken into account into the prediction.

Response 4: Thanks for raising this issue. Yes, in an ideal scenario, an optimum risk prediction model will include time varying risk factors or include factor of time. This would likely be the case if we design a study and followed it over time. However, we are using multiple cohorts with different baseline dates, and not all the time-varying risk factors are available in the longitudinal follow-up. We therefore aimed to develop a risk prediction model targeting mid-life (aged 45 to 65 years) and a risk prediction model for late-life (65-years onward). In addition, the available published risk prediction models for dementia are based on measurement of risk factors at baseline (3, 4). We have added some text related to this in the protocol paper.

Comment 5. Page 10, line 53- candidate predictors

It would be great if type of variables can be specified-either continuous, ordinal or binary. categorical

Response 5.Thanks, we have now specified the type of variables in Table 4.

Comment 6. Page 11, line 30: Sample size

Please provide evidence to support the sample size calculation used for the model. Besides, is there any number of data used for model development and model validation? How authors determine this.

Response 6: Thanks for your comment. The main objective of the protocol is to develop and validate a risk prediction model for NCDs including dementia that would determine individuals will higher or lower risk so that they can modify the risk factor if they are in higher risk group. In this regards, we have estimated cohorts sizes based on dementia prevalence's (which is the lowest among all NCDs) as shown in the sample size calculation on page 10. The sample size calculation for Fine and Gray model is not relevant here as we are not primarily interested in estimating survival probability (here in our case, the probability of the time to get any of the NCDs given the risk factors of any of the event). Regarding the model development and validation, in the manuscript page11, line 16 we have stated that 65% of the data will be used for model building and 35% for the validation. This is common practice and we have added a reference.

Comment 7. Page 12, line 21

If this is the case, the current work may not produce validated tool as stated in the title. Internal validation is a must procedure to develop a model.

Response 7: Yes, we will be using 35% data to validate the model. Sorry for the confusion with the text. We have clarified the text in the analysis plan section.

Comment 8. Page 12, line 28

Any specific duration of time to the event e.g., dementia. Will you specify the risk based on different duration?

Response 8: We have briefly mentioned this in model specification on page 13, line 50-52, "As we plan to evaluate the competing risk model in individual datasets, we will not be able to create 10-year risk prediction estimates." Hence we will not be able to calculate dementia risk based on different duration.

Comment 9. Page 14: line 3

What type of stress measures will be incorporated into the model?

Response 9: Stress was identified as one of the candidate predictors to be included in the model. Self reported stress measures such as one's ability to control important things in life and perceived stress scale questions are some of the stress measures included in the studies. Types of stress may not be available in the datasets.

Comment 10. Page 15: Any specific method used for this meta regression. Will you use machine learning approach? As meta regression applying conventional methods- there are few factors need to be considered and explained such multicollinearity of the data and cofounding factors that may affect the prediction. Authors may need to explain how they will take this into account.

Response 10: Thanks for pointing this. We think there is a typo in the text. We have amended "meta regression" to meta-analysis of the regression coefficients obtained across all cohorts.

Comment 11. How the internal validation will be performed?

Response 11: Data will be pooled from the ten cohort studies and 65% of the pooled data will be used for training while the remaining 35% of the data will be used for as test data for internal validation. Area under the curve and Harrell's c-statistics will be used for evaluating the prediction model. This has been mentioned on Page 11 under the analysis plan.

Comment 12. Any specific method used for this survival model development?

Response 12: Thanks. We have mentioned in our analysis plan on page 11, line 17 that we will be using Fine and Gray sub distribution model and added a reference to the specific model.

Comment14. Page 16, line 47 What would be the difference between the two tools and why it should be different?

Response 14: The algorithms to predict the four outcomes will be the same for the clinicians and the public/researchers' assessment tool. The only difference will be the length of the questionnaire. As clinicians have limited consultation time, their questionnaire will be a shorter questionnaire as compared to the public one. We also plan to collect data on relatively new risk factors via the public tool so that we are able to update the tool appropriately going forward if additional key risk factors emerge. We have added this information in the manuscript to make it clear.

References

1. Anstey KJ, Cherbuin N, Herath PM. Development of a New Method for Assessing Global Risk of Alzheimer's Disease for Use in Population Health Approaches to Prevention. *Prevention science* 2013;14(4):411-21;10.1007/s11121-012-0313-2.
2. Chen L, Magliano DJ, Balkau B, et al. AUSDRISK: an Australian Type 2 Diabetes Risk Assessment Tool based on demographic, lifestyle and simple anthropometric measures. *Medical Journal of Australia*. 2010;192(4):197-202.
3. Huque MH, Kootar S, Eramudugolla R, et al. A comparative analysis of the CogDrisk, ANU-ADRI, CAIDE and LIBRA risk scores for predicting dementia in three population-based cohorts (in-press). *JAMA Network open*. 2023.
4. Kootar S, Huque M, Eramudugolla R, et al. Validation of the CogDrisk Instrument as Predictive of Dementia in Four General Community-Dwelling Populations. *The Journal of Prevention of Alzheimer's Disease*. 2023:1-10.